

# Glycerol-based extracts of *Clitoria ternatea* (Butterfly Pea Flower) with enhanced antioxidant potential

Lai Ti Gew[1,2], Waye Juin Teoh[1], Li Lin Lein[1,2,3], Min Wen Lim[1], Patrick Cognet[4] and Mohamed Kheireddine Aroua[2,5,6]

[1] Department of Biological Sciences, Sunway University, Petaling Jaya, Selangor, Malaysia
[2] Sunway Materials Smart Science and Engineering (SMS2E) Research Cluster, Sunway University, Petaling Jaya, Selangor, Malaysia
[3] School of Biosciences, Faculty of Health & Medical Sciences, Taylor's University, Petaling Jaya, Selangor, Malaysia
[4] Institut National Polytechnique de Toulouse, Toulouse, None Selected, France
[5] Department of Engineering, Lancaster University, Lancaster, United Kingdom
[6] Centre for Carbon Dioxide Capture and Utilization (CCDCU), School of Engineering and Technology, Sunway University, Petaling Jaya, Selangor, Malaysia

Corresponding author
Lai Ti Gew, janeg@sunway.edu.my

## ABSTRACT

The butterfly pea flower (*Clitoria ternatea*) is a plant species that is commonly used in culinary products, as it adds a natural purplish-blue tint to dishes without artificial food colourings and is rich in antioxidants. In this study, glycerol was employed as an extraction solvent for the extraction of phenolic compounds from *C. ternatea*. Several studies have proven glycerol is an ideal green solvent to replace conventional solvents such as ethanol and methanol due to its ability to change the water polarity, thereby improving the extraction of bioactive compounds and recovering the polyphenols from natural products. We systematically reviewed the phytochemical content and antioxidant properties of aqueous, ethanol and methanol extracts of *C. ternatea* as a comparison to our study. Our results show that glycerol extract (GE) and glycerol/water extract (GWE) have demonstrated high phenolic and flavonoid profiles as compared to ethanol extract (EE) and water extract (WE). This study suggests glycerol as a promising extraction medium to extract higher concentrations of phytochemical contents from *C. ternatea*. It could be used as a natural source of antioxidant boosters, particularly in food preparation and cosmeceutical product development.

## INTRODUCTION

The butterfly pea flower (*Clitoria ternatea*) is a common plant species that has been cultivated in many Asia countries. It is a perennial twinning herbaceous plant that belongs to the family Fabaceae and sub-family Papilionaceae. The plant is believed to have originated from Asia and was later brought to South and Central America. In the 17th century, the butterfly pea plant was distributed widely around China and India, and later it was distributed to Europe and tropical countries (*Pwee, 2016*). Nowadays, this plant can

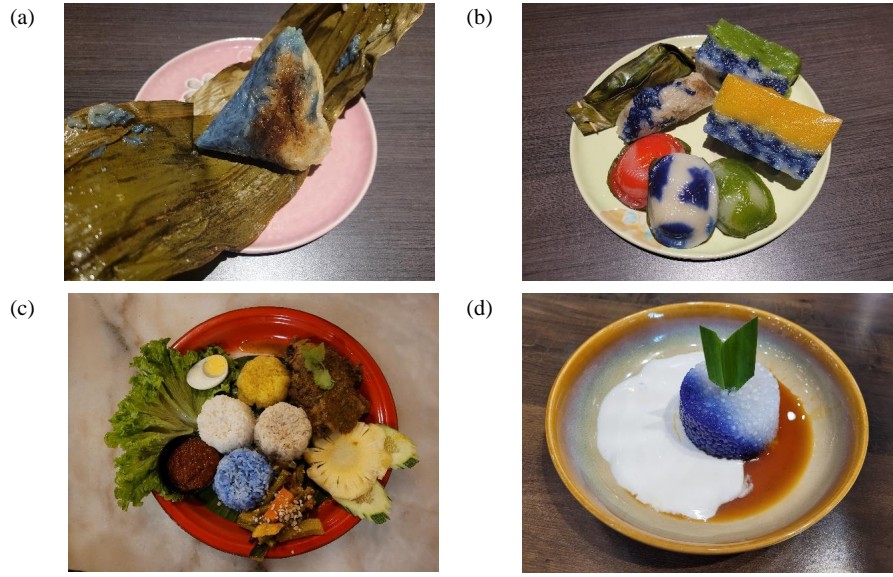

**Figure 1** **Delicacies containing natural colouring from *C. ternatea* flowers extracts.** (A) Nyonya chang (Peranakan rice dumplings), (B) kuih including ang ku, pulut tekan (a Peranakan glutinous rice dessert), Rempeh udang (a Peranakan glutinous rice with shrimp paste), (C) nasi kerabu (a local rice cuisine from Kelantan), and (D) sago (a dessert). Pictures were taken by Lai Ti Gew during her trip in Melaka, Malaysia on 15 January 2022.

be found in most Southeast Asian countries including Thailand, Myanmar, Malaysia, and Vietnam.

*C. ternatea* flowers are known as 'Bunga Telang' or 'Bunga Biru' by Malaysians according to Malaysia Biodiversity Information System (MyBIS) (retrieved from https://www.mybis.gov.my/sp/36158 on 3 January 2022). These flowers are commonly used in many areas such as the culinary arts and medicinal applications due to their functionality and practicality. In Malaysia, these edible flowers are used as natural food colouring in nasi kerabu (a local rice cuisine from Kelantan, Malaysia), nyonya zhang (Peranakan rice dumplings) and kuih pulut tekan (a Peranakan glutinous rice dessert originated from Melaka, Malaysia) (Fig. 1). The flowers add a natural purplish-blue tint to these delicacies without the need to use artificial food colourings. In other countries like India and Burma, butterfly pea flowers are consumed as vegetables, used as garnish in salads, used as ingredients in cooking, or fried in batter as a snack. In Thailand, these flowers are served alongside pandan-flavoured syrup, honey, or lime juice as a refreshing blue beverage. This butterfly pea flower tea is rich in antioxidants that can strengthen the body's immune system. The antioxidants in the tea also aid in fighting wrinkles and preventing the skin from ageing. Hence, the various health benefits of *C. ternatea* make it a wonderful addition to many foods and beverages.

Besides culinary uses, the whole *C. ternatea* plant contains many health purposes. Each component of the plant has been utilised in various medicinal usages. For example, the seeds and leaves of *C. ternatea* are reported to be used as poultices to treat swollen joints

(*Pwee, 2016*). The juice of the flowers is revealed to treat a wide range of ailments like insect bites, inflamed eyes, and skin disorders. The roots of the plant are also used to cure several health conditions such as infections, asthma, body aches and pulmonary tuberculosis (*Zingare et al., 2013*). Moreover, the *C. ternatea* plant has been revealed to contain various biological properties like antioxidant, antimicrobial, antidiabetic, antihyperlipidemic, and hepatoprotective activities (*Escher et al., 2020*; *Lakshan et al., 2019*). Interestingly, a group of researchers in Thailand developed a novel formula for sponge cake by replacing cake flour with spray-dried powder of *C. ternatea* flower petal extract ranging from 5 to 20% (w/w) in the formulations. Total phenolic and total flavonoid content, as well as the antioxidant assays, namely, 2,2-diphenyl-1-picrylhydrazyl (DPPH) free radical scavenging and ferric reducing antioxidant power (FRAP) assays of water extract of sponge cakes, were performed. They found that the replacement of flower extract increases the polyphenol content as well as the antioxidant properties (*Pasukamonset et al., 2018*). Thus, *C. ternatea* flower extract is a potential source to be employed in the development of functional food.

Various research on *C. ternatea* flower extracts are commonly prepared using water, and organic solvents such as ethanol, methanol, ethyl acetate, and acetone (Table 1). The use of organic solvents in the isolation and purification of phenolic compounds from natural products has been increasingly exploited due to its significant antioxidant potential and substantial health benefits. Natural polyphenols are the most widely distributed group of secondary plant metabolites that are generally responsible for protection from ultraviolet radiation as well as resistance against pathogens (*Ganesan & Xu, 2017*). In the food industry, polyphenols play an important role in the colouring, astringency and bitterness of foods. Furthermore, the strong antioxidant activity of polyphenols offers substantial health benefits such as protection against the development of diseases and cancers (*Maqsood, Benjakul & Shahidi, 2013*). Moreover, the antioxidant capacity of plants depends on the chemical composition of phenolic compounds, which makes them highly effective antioxidant agents with high radical scavenging activities (*Álvarez et al., 2016*). Even though organic solvents are highly effective solvents and possess a wide range of advantages, their potential to cause adverse effects on human health and the environment are the major concerns in the food processing industry (*Pena-Pereira, Kloskowski & Namieśnik, 2015*). In view of all shortcomings, it is necessary to develop a method by using effective green solvents that could perform green extraction and give high extraction yield, high selectivity and high purity, as well as zero-waste biorefinery (*Chemat, Vian & Cravotto, 2012*). Thus, their replacement with eco-friendly alternatives with greener technologies would involve remarkable advances that could meet both technologies and economic demands.

A significant amount of glycerol produced during the biofuel manufacturing process has been directed to the exploration of the use of this major by-product. In general, for every 100 pounds of biodiesel produced, approximately 10 pounds of crude glycerol are produced. An increasing amount of glycerol production has led to waste disposable environmental issues as it is classified as an unrefined raw material and the use of glycerol is limited. There is a need to refine it and produce high-value products from glycerol, which may help consolidate the sustainability of the biofuel production market. Repurposing this major

**Table 1  The phytochemical content and antioxidant properties of aqueous, ethanol and methanol extracts of *C. Ternatea*.**

| Solvent | Sample preparation | Phytochemical content | Antioxidant assays | Reference |
|---|---|---|---|---|
| Water | Macerating 200 mg dry flower with 100 mL distilled water | TPC: 1.9 mg GAE/g | DPPH radical activity ($IC_{50}$): 1,000 $\mu$g/mL Std: Trolox | *Kamkaen & Wilkinson (2009)* |
| | 200 g dried flowers extracted with 1 L distilled water at 90 °C for 2 h Spray drying process | TPC: 233.33 ± 17.64 mg GAE/g TFC: 78.28 ± 1.47 mg QE/g | n/a | *Adisakwattana et al. (2012)* |
| | Dried plant material was coarsely powdered, and the filtrate was collected after 3 h of hot extraction in distilled water | TPC: 1.3805 ± 0.0303 mg GAE/g TFC: 55.0 ± 3.0 $\mu$M rutin eq/g | ABTS (IC50): 87.29 ± 0.8 $\mu$g/mL | *Mehla et al. (2013)* |
| | 0.5 g dried plant powder extracted using 25 mL deionised water, then incubated in dark at room temperature for 1 h, with occasional agitation | TPC: 20.7 ± 0.1 mg GAE/g | % Scavenging activity per sample extract concentration ($\mu$g/mL): DPPH (at 25$\mu$g/mL): 390.67 ± 2.309% DPPH (at 50 $\mu$g/mL): 401.33 ± 3.055% DPPH (at 100 $\mu$g/mL): 449.33 ± 2.309% DPPH (at 125 $\mu$g/mL): 490.67 ± 4.619% DPPH (at 150 $\mu$g/mL): 506.67 ± 2.309% | *Rabeta & An Nabil (2013)* |
| | 3 kg of dried crushed flower in 10 L distilled water at 95–100 °C for 30 min | n/a | DPPH (IC50): 84.15 ± 1.50 $\mu$g/mL FRAP: 330 ± 10 mmol AAE/g (Ascorbic acid as standard) | *Iamsaard et al. (2014)* |
| | 0.5 kg dried flower petals were boiled in 3 L of distilled water for 2 h | TPC: 53.00 ± 0.34 mg GAE/g TFC: 11.20 ± 0.33 mg CE/g TAC: 1.46 ± 0.04 mg CGE/g (Std: Cyanidin-3-glucoside) | DPPH (IC50): 470 ± 10 $\mu$g/mL (or 0.47 ± 0.01 mg/mL) | *Phrueksanan, Yibchok-anun & Adisakwattana (2014)* |
| | 300 g dry flower extracted with 1 L distilled water at 95 °C for 2 h. | TPC: 53 ± 0.34 mg GAE/g TFC: 11.2 ± 0.33 mg CE/g TAC: 1.46 ± 0.04 mg CGE/g | DPPH (IC50): 467 ± 5 $\mu$g/mL ABTS: 0.168 ± 0.001 mg TE/mg (Trolox as standard) FRAP: 0.379 ± 0.009 mmol $FeSO_4$E/mg ($FeSO_4$ as standard) | *Chayaratanasin et al. (2015)* |
| | Raw material stirred in distilled water for 10 min at 100 °C, ratio of plant parts to water is 1:4 (v/v) | TPC: 76.90 mg GAE/g TFC: 16.19 mg QE/g TAC: 6.93 mg CGE/g | DPPH (EC50): 760 ± 30 $\mu$g/mL ABTS: 4.16 ± 0.12 $\mu$M TEAC/g FRAP: 10.91 ± 0.60 mM TEAC/g (Trolox as standard) | *Siti Azima, Noriham & Manshoor (2017)* |

Gew et al. (2024), *PeerJ Analytical Chemistry*, DOI 10.7717/peerj-achem.30

**Table 1** (*continued*)

| Solvent | Sample preparation | Phytochemical content | Antioxidant assays | Reference |
|---|---|---|---|---|
| | 50 g powdered samples extracted in 500 mL freshly boiled deionised water for 30 min, followed by sonication for 15 min | TPC: 38.5 mg GAE/g | DPPH (IC50): 195.5 µg/mL<br>ABTS (IC50): 42.9 µg/mL | *Zakaria (2019)* |
| | 0.2 g dried ground petals mixed in 4 mL distilled water, soaked for 0, 6, 12 and 24 h | TPC (0 h): 13.8 ± 0.5 mg GAE/g<br>TPC (6 h): 13.5 ± 1.0 mg GAE/g<br>TPC (12 h): 13.9 ± 0.0 mg GAE/g<br>TPC (24 h): 12.2 ± 2.1 mg GAE/g<br>TAC (0 h): 58.2 ± 9.6 mg/L<br>TAC (6 h): 51.8 ± 6.9 mg/L<br>TAC (12 h): 49.0 ± 5.6 mg/L<br>TAC (24 h): 39.9 ± 11.5 mg/L | DPPH (0 h): 10.9 ± 0.4 mM TE/g<br>DPPH (6 h): 11.7 ± 0.4 mM TE/g<br>DPPH (12 h): 11.1 ± 0.4 mM TE/g<br>DPPH (24 h): 9.45 ± 2.1 mM TE/g | *López Prado et al. (2019)* |
| | Dried flowers were powdered using electric blender<br>With ultrasound (US): solid to liquid ratio 1 g:15 mL, 50 °C, 150 min, 70% amplitude, 240 W.<br>Without ultrasound (Agitation/AGE): solid: liquid ratio 1 g:15 mL, 50 °C, 150 min. | TPC (US): 87.00 ± 1.25 mg GAE/g<br>TFC (US): 29.00 ± 1.40 mg QE/g<br>TPC (AGE): 72.00 ± 1.90 mg GAE/g (or 7200 mg GAE/100 g)<br>TFC (AGE): 25.00 ± 1.06 mg QE/g | DPPH (US): 0.931 ± 0.017 mg TE/g (or 931.46 ± 16.91 µg TE/g)<br>DPPH (AGE): 0.764 ± 0.023 mg TE/g (or 764.32 ± 23.41 µg TE/g)<br>ABTS (US): 13.488 ± 0.686 mg TE/g (or 13,488 ± 685.801 µg TE/g ABTS (AGE): 11.720 ± 0.218 mg TE/g (or 11,720.33 ± 217.7910 µg TE/g)<br>FRAP (US): 5.834 mg TE/g (or 5,834.59 µg TE/g)<br>FRAP (AGE): 4.195 mg TE/g (or 4,195.29 µg TE/g) | *Mehmood et al. (2019)* |
| | 3 g of powdered dry flower in 1 L distilled water, 59.6 °C, 37 min | TPC: 26.72 ± 2.17 mg GAE/g (or 80.17 ± 6.51 mg GAE/L)<br>TFC: 14.25 ± 0.58 mg QE/g (or 42.75 ± 1.74 mg QE/L) | DPPH (IC50): 725.52 ± 23.52 µg/mL (or 241.84 ± 7.84 µL/mL)<br>ABTS (IC50): 104.13 ± 5.40 µg/mL (or 34.71 ± 1.80 µL/mL)<br>FRAP: 5.13 ± 0.54 mg TE/g (or 15.39 ± 1.63 mg TE/L) (Trolox as standard for FRAP) | *Lakshan et al. (2019)* |
| | 1 g of dry flower in 25 ml of distilled water and agitated using an orbital shaker for 2 h at 50 °C | TPC: 0.273 ± 0.0196 mg GAE/g | DPPH: 69.22 ± 1.60% inhibition<br>ABTS: 364.27 ± 7.14 µg TE/g<br>FRAP: 28.21 ± 2.39 mg Fe (II)/g | *Wong & Tan (2020)* |

**Table 1** (*continued*)

| Solvent | Sample preparation | Phytochemical content | Antioxidant assays | Reference |
|---|---|---|---|---|
| | Ground dried petals mixed in distilled water Liquid/solid ratio (5.0, 7.5 and 10.0 ml/g) mixed well using ultrasound Temperature: 40, 60 and 80 °C Time: 30, 45 and 60 min | TPC ranging from 3.09 ± 0.02 to 6.67 ± 0.01 mg GAE/g TAC ranging from 1.05 ± 0.06 to 2.19 ± 0.14 mg cy-3-glu/g | DPPH ranging from 40.44 ± 3.22 to 63.24 ± 1.17% | *Salacheep et al. (2020)* |
| | 150 g dried flower in 1,500 mL distilled water then micro-encapsulate with 2 to 10% of gum Arabic. | TPC (2%): 38.09 ± 2.73 mg GAE/g TPC (4%): 35.85 ± 3.67 mg GAE/g TPC (6%): 30.85 ± 3.25 mg GAE/g TPC (8%): 32.69 ± 2.77 mg GAE/g TPC (10%): 35.65 ± 2.57 mg GAE/g TFC (2%): 4.60 ± 0.23 mg QE/g TFC (4%): 5.35 ± 0.17 mg QE/g TFC (6%): 6.90 ± 0.34 mg QE/g TFC (8%): 6.73 ± 0.25 mg QE/g TFC (10%): 5.93 ± 0.65 mg QE/g | DPPH (2%): 49.53 ± 8.94% DPPH (4%): 56.92 ± 0.81% DPPH 6%): 67.32 ± 5.55% DPPH (8%): 70.25 ± 9.11% DPPH (10%): 60.25 ± 9.11% | *Zainol et al. (2020)* |
| | Dried powdered flowers were added with aquadest, heated to 50 °C Solid in liquid (g/mL): 1 g in 20 ml of distilled water (1:20), 1 g in 50 ml of distilled water (1:50) Time (min): 90 and 150 min pH: 1 and 7. | TPC (1:20, 90, 1): 57.92 mg GAE/ mg TPC (1:20, 90, 7): 65.98 mg GAE/ mg TPC (1:20, 150, 1): 59.31 mg GAE/ mg TPC (1:20, 150, 7): 66.72 mg GAE/ mg TPC 1:50, 90, 1): 17.33 mg GAE/ mg TPC (1:50, 90, 7): 94.04 mg GAE/ mg TPC (1:50, 150, 1): 7.88 mg GAE/ mg TPC (1:50, 150, 7): 56.02 mg GAE/mg TAC (1:20, 90, 1): 1206.77 mg/L TAC (1:20, 90, 7): 362.92 mg/L TAC (1:20, 150, 1): 961.86 mg/L TAC (1:20, 150, 7): 811.57 mg/L TAC (1:50, 90, 1): 783.18 mg/L TAC 1:50, 90, 7): 478.14 mg/L TAC (1:50, 150, 1): 691.33 mg/L TAC (1:50, 150, 7): 450.31 mg/L | n/a | *Aditiyarini & Iswuryani (2021)* |
| | 0.125 g dehydrated ground petals in 25 ml ultrapure water Temperature ranging from 11.7 to 68.3 °C Time ranging from 8.78 to 51.21 min | TPC ranging from 5.96 ± 0.08 to 6.92 ± 0.14 mg GAE/g TFC ranging from 5.46 ± 0.01 to 6.21 ± 0.01 mg QE/g TAC ranging from 3.53 ± 0.01 to 3.96 ± 0.01 mg mg CGE/g | Percentage inhibition of DPPH radical ranging from 55 ± 1 to 65 ± 1% FRAP ranging 12.43 ± 0.69 to 15.74 ± 0.15 mg AAE/g | *Escher et al. (2020)* |

Gew et al. (2024), *PeerJ Analytical Chemistry*, DOI 10.7717/peerj-achem.30

**Table 1** (*continued*)

| Solvent | Sample preparation | Phytochemical content | Antioxidant assays | Reference |
|---|---|---|---|---|
| Ethanol | Macerating 200 mg dry flower with 100 mL 95% ethanol | n/a | DPPH radical activity (IC$_{50}$): 4,000 µg/mL Std: Trolox | *Kamkaen & Wilkinson (2009)* |
| | Microwave assisted extraction (MAE) of powdered samples in 95% ethanol Liquid to solid ratio ranging from 11 to 29 g/mL Temperature ranging from 32 to 68 °C Time ranging from 11 to 29 min | TAC ranging from 0.1459 mg/g to 0.4571 mg/g | n/a | *Izirwan et al. (2020)* |
| | 0.1 g freeze-dried flower in 10 mL 37% ethanol for 90 min at 45 °C | TPC: 41.17 ± 0.5 mg GAE/g TFC: 187.05 ± 3.18 mg QE/g TAC: 28.60 ± 0.04 mg/L | DPPH scavenging activity: 63.53 ± 0.95% | *Jaafar, Ramli & Salleh (2020)* |
| | 5 g of plant material extracted using 95 g of water and ethanol mixture (80:20) with ultrasound assistance | TPC: 15.62 ± 0.14 mg GAE/g TFC: 7.26 ± 0.12 mg QE/g | DPPH (500 µg/mL): 20% DPPH (1,000 µg/mL): 39% | *Zagórska-Dziok et al. (2021)* |
| | 100 g powdered sample macerated with 500 mL 70% ethanol for 72 h at room temperature with occasional shaking | TPC: 53.0 mg GAE/g | DPPH (IC50): 188.9 µg/mL ABTS (IC50): 37.8 µg/mL | *Zakaria (2019)* |
| | Grounded dried sample was extracted using 60% ethanol in shaking water bath at 60 °C with a speed of 100 rpm for 2 hrs Sample to solvent ratio was 1:8 | TPC (750 nm): 28.75 ± 1.215 mg GAE/g TPC (280 nm): 102.37 ± 1.063 mg GAE/g TFC: 35.73 ± 0.978 mg QE/g TAC: 2.88 ± 0.408 mg ME/g (Malvidin as standard) TAC: 2.72 ± 0.386 mg CE/g (Cyanidin as standard) | DPPH: 0.733 ± 0.002 µMol Ascorbic acid/L DPPH: 0.55 ± 0.009 µMol Trolox/g DPPH inhibition: 42.40 ± 0.370% DPPH (IC50): 2.77 ± 0.020 mg/g ABTS: 5.90 ± 0.080 µMol Ascorbic acid/L ABTS: 5.84 ± 0.080 µMol Trolox/g ABTS inhibition: 29.16 ± 0.425% ABTS (IC50): 10.23 ± 0.186 mg/g | *Tuan Putra et al. (2021)* |
| | 1 g crushed fresh whole flowers extracted in 20 mL 95% ethanol with microwave assistance (400 W) for 3 min For every 15 s, the oven is stopped for 15 s to prevent overheating | TPC: 26.90 ± 1.12 mg GAE/g TAC: 254.63 mg/kg | DPPH scavenging activity: 27.78 ± 0.12 mg TE/g | *Saejung, Don-In & Chim-sook (2021)* |

Gew et al. (2024), *PeerJ Analytical Chemistry*, DOI 10.7717/peerj-achem.30

**Table 1** (*continued*)

| Solvent | Sample preparation | Phytochemical content | Antioxidant assays | Reference |
|---|---|---|---|---|
| Methanol | 5 g sample mixed with 50 mL methanol/HCl (100:1, v/v), containing 2% tert-butylhydroquinone | TPC (Soluble): Around 60 mg GAE/g<br>TPC (Bound): Around 60 mg GAE/g<br>TFC (Soluble): Around 15 mg RE/g<br>TFC (Bound): Around 5 mg RE/g<br>(Rutin as standard) | DPPH radical inhibition (Soluble): 32.7 ± 2.75%<br>DPPH radical inhibition (Bound): 17.59 ± 2.91% | *Kaisoon et al. (2011)* |
| | 100 g dried powdered sample soaked in 300 mL 100% methanol for 4 days at room temperature, stirred occasionally | TPC: 105.40 ± 2.47 mg GAE/g<br>TFC: 72.21 ± 0.05 mg CE/g | DPPH radical scavenging activity: 68.9%<br>DPPH (IC50): 327 $\mu$g/mL | *Nithianantham et al. (2013)* |
| | Dried plant material was coarsely powdered, and the filtrate was collected after 3 h of hot extraction in 50% methanol | TPC: 1.4237 ± 0.0201 mg GAE/g<br>TFC: 78.0 ± 3.0 $\mu$M rutin eq/g | ABTS (IC50): 56.62 ± 1.7 $\mu$g/mL | *Mehla et al. (2013)* |
| | 5 g dried powder added to 100 mL of 70% methanol, incubated overnight in orbital shakers | TPC: 61.7 ± 0.2 mg GAE/g | % Scavenging activity per sample extract concentration ($\mu$g/mL):<br>DPPH (25 $\mu$g/mL): 32.67 ± 1.155%<br>DPPH (50 $\mu$g/mL): 353.33 ± 3.055%<br>DPPH (100 $\mu$g/mL): 411.33 ± 1.155%<br>DPPH (125 $\mu$g/mL): 422.67 ± 3.055%<br>DPPH (150 $\mu$g/mL): 401.33 ± 2.309% | *Rabeta & An Nabil (2013)* |
| | 750 g air dried flowers were extracted three times with 2,000 mL of 95% methanol (4*500 mL) at room temperature (30 ± 2 °C) | n/a | DPPH free radical scavenging effect per concentration (mg/mL):<br>1.0 mg/mL: 85.27 ± 0.02%<br>0.5 mg/mL: 77.68 ± 0.60%<br>0.25 mg/mL: 74.51 ± 0.01%<br>0.125 mg/mL: 56.12 ± 0.05%<br>IC50: 95.30 ± 0.10 $\mu$g/mL | *Rajamanickam, Kalaivanan & Sivagnanam (2015)* |
| | 0.2 g dried ground petals were mixed in 4 mL 100% methanol, soaked for 0, 6, 12 and 24 h | TPC (0 h): 5.72 ± 0.7 mg GAE/g<br>TPC (6 h): 5.89 ± 0.2 mg GAE/g<br>TPC (12 h): 7.36 ± 0.4 mg GAE/g<br>TPC (24 h): 13.7 ± 1.8 mg GAE/g<br>TAC (0 h): 49.3 ± 5.4 mg/L<br>TAC (6 h): 51.2 ± 3.2 mg/L<br>TAC (12 h): 52.1 ± 4.8 mg/L<br>TAC (24 h): 94.1 ± 10.6 mg/L | DPPH (0 h): 6.36 ± 0.3 mM TE/g<br>DPPH (6 h): 6.99 ± 0.5 mM TE/g<br>DPPH: (12 h): 11.7 ± 1.3 mM TE/g<br>DPPH (24 h): 8.81 ± 0.6 mM TE/g | *López Prado et al. (2019)* |

**Table 1** (*continued*)

| Solvent | Sample preparation | Phytochemical content | Antioxidant assays | Reference |
|---|---|---|---|---|
| | 0.2 g dried ground petals mixed in 2mL distilled water and 2 mL methanol (1:1), soaked for 0, 6, 12 and 24 h. | TPC (0 h): 14.1 ± 0.6 mg GAE/g<br>TPC (6 h): 11.7 ± 0.7 mg GAE/g<br>TPC (12 h): 12.6 ± 0.3 mg GAE/g<br>TPC (24 h): 14.5 ± 1.4 mg GAE/g<br>TAC (0 h): 60.0 ± 4.2 mg/L<br>TAC (6 h): 63.9 ± 6.1 mg/L<br>TAC (12 h): 59.9 ± 2.7 mg/L<br>TAC (24 h): 64.8 ± 8.26 mg/L | DPPH (0 h): 0: 11.4 ± 0.7 mM TE/g<br>DPPH (6 h):12.2 ± 1.0 mM TE/g<br>DPPH: (12 h): 11.2 ± 0.6 mM TE/g<br>DPPH (24 h): 11.3 ± 1.1 mM TE/g | *López Prado et al. (2019)* |
| | 100 mg fresh chopped samples extracted in 5 mL of 99.95% methanol | TPC ranging from 44.7 ± 9 to 78.7 ± 30 mg GAE/100 g<br>TAC ranging from 0.0 ± 0 to 73.1 ± 3 mg/100 g | DPPH ranging from 4.5 ± 6 to 67.7 ± 24 mg TEAC/g<br>FRAP ranging from 15.4 ± 2 to 27.8 mg TEAC/g | *Havananda & Luengwilai (2019)* |
| Ethyl acetate | | n/a | DPPH free radical scavenging effect per concentration (mg/mL):<br>1.0 mg/mL: 88.79 ± 0.20%<br>0.5 mg/mL: 83.91 ± 0.01%<br>0.25 mg/mL: 77.88 ± 0.50%<br>0.125 mg/mL: 66.05 ± 0.04%<br>IC50: 107.42 ± 0.02 µg/mL | *Rajamanickam, Kalaivanan & Sivagnanam (2015)* |
| Chloroform | The methanolic extract from *Rajamanickam, Kalaivanan & Sivagnanam (2015)* was suspended in hot water (1,000 mL), and then partitioned with chloroform and ethyl acetate respectively. | n/a | DPPH free radical scavenging effect per concentration (mg/mL):<br>1.0 mg/mL: 92.46 ± 0.05%<br>0.5 mg/mL: 88.42 ± 0.04%<br>0.25 mg/mL: 80.94 ± 0.01%<br>0.125 mg/mL: 76.79 ± 0.02%<br>IC50: 132.50 ± 0.06 µg/mL | *Rajamanickam, Kalaivanan & Sivagnanam (2015)* |
| Citrate buffer | 1:4 (v/v) of raw (fresh?) flower to 100 mM citrate buffer ratio, pH 3.0, 100 °C for 10 min | TAC: 16.07 ± 0.02 mg CE/g | DPPH (EC50): 490 ± 10 µg/mL | *Siti Azima, Noriham & Manshoor (2017)* |

by-product will increase the profitability and thus sustainability of the biofuel production market. Furthermore, better management of waste (in this case glycerol) will reduce carbon emissions into the environment and increase the capacity to effectively plan and manage climate action. In the field of green chemistry, some researchers demonstrated the feasibility of using glycerol as a solvent in organic synthesis, enhancing reaction selectivity, facilitating the reaction products and separation, and as a biocatalyst. Glycerol is water soluble, miscible with ethanol (polarity index = 4.3) and slightly soluble in ethyl ether (polarity index = 2.8). The wide range of polarity index makes it an excellent solvent in extracting polar and slightly nonpolar compounds from natural products (*Gu & Jérôme, 2010*). Furthermore, the chances of extracting the nonpolar plant steroid may be low by using glycerol. Thus, the use of glycerol could be a sustainable approach to solving the problem of large amounts of solvent usage in natural products. In the food industry, glycerol can act as a solvent, sweetener, humectant and much more. Furthermore, glycerol is suitable for food preparation because it is digestible, non-toxic, safe for human consumption and enhances the flavour and odour of the food product. For instance, *Candida magnolia*, an osmophilic yeast, was used to convert crude glycerol into mannitol, which is a type of sugar alcohol commonly used to replace sugar in food preparation (*Azelee et al., 2019*). Besides that, glycerol can be converted into glycerol monolaurate through an esterification process. Glycerol monolaurate serves as an important preservative and surfactant in the food industry. Humectants are used in food preparations to increase the water-holding capacity of the food products. Therefore, the stability and the texture of the food product can be improved by adding humectants. Glycerol is classified as an effective humectant polyol. It contains strong moisturization characteristics in food due to the presence of hydroxyl groups, which allows the glycerol to attach and retain water. In addition, when glycerol is added to the food product, it is believed to reduce the growth of foodborne pathogens by reducing water content (*Finn et al., 2015*). Furthermore, glycerol has been utilised in meat products to improve meat quality, emulsifying capability, and water-binding ability. It is also added into meat jerky products to lower water content and decrease protein aggregation. It is worth noting that the toxicity of glycerol is relatively low when ingested by humans. The reported value of toxic dose low (TDLo) is 1,428 mg/kg orally (retrieved from https://pubchem.ncbi.nlm.nih.gov/compound/Glycerol#section=Toxicity, accessed on 3 January 2022).

However, limited published work showed that aqueous glycerol was successfully used to extract polyphenolic antioxidants from natural products, such as olive leaves (*Apostolakis, Grigorakis & Makris, 2014*), *Hypericum perforatum* (*Karakashov et al., 2015*), Artemisia species (*Shehata et al., 2015*), coffee (*Michail et al., 2016*), rice bran (*Aalim et al., 2019*) and lotus (*Huang et al., 2019*). Results demonstrated that the incorporation of glycerol as one of the extraction solvents gives a high extraction yield and high selectivity. The use of glycerol is hypothesised as a sustainable solvent that could be more effective and safer than organic solvents in the extraction of bioactive compounds.

In this study, glycerol was employed for the extraction of phenolic compounds from *C. ternatea*. With respect to its cost-effective and non-toxicity, several studies have proven glycerol is an ideal green solvent to replace conventional solvents such as ethanol and

**Table 2   Search string for article search using SCOPUS database and PubMed.**

("blue pea flower" OR "butterfly pea flower" OR "Clitoria ternatea")
AND ("extract" OR "water extract" OR "aqueous extract" OR "ethanol
extract" OR "methanol extract") AND ("total phenolic content" OR "total
flavonoid content" OR "2,2-diphenyl-1-picrylhydrazyl")

methanol due to its ability to change the water polarity, thereby improving the extraction of bioactive compounds and recovering the polyphenols from natural products. To evaluate the efficiency of glycerol as an extraction solvent, the total phenolic content, total flavonoid content and 2,2-diphenyl-1-picryl-hydrazyl-hydrate (DPPH) radical scavenging activity of *C. ternatea* were investigated. Additionally, we also reviewed systematically the phytochemical content and antioxidant properties of water, ethanol and methanol extracts of *C. ternatea* as a comparison to our study.

## LITERATURE REVIEW

### Search methodology

The articles search was performed using the search string shown in Table 2 through the SCOPUS database and the National Centre for Biotechnology Information (NCBI) (PubMed and PubMed Central database) in September 2021. A total of 206 primary research articles were retrieved from SCOPUS, six from PubMed and 76 from PubMed Central. It is worth mentioning that we do not set any limitations on the timeline. Out of the 206, two in-press articles and two non-English articles were excluded from the SCOPUS list. Only six articles from PubMed and PubMed Central were included upon cross-checking any redundancies in the overlapping lists. Full texts were downloaded from publisher sites such as Elsevier, Springer and MDPI. A thorough examination of the title, abstract and main contents of all articles was done to ensure that they were relevant to our topic. A total of 27 articles with full texts were included in this literature review section (Table 1).

### Extraction methods

A systematic tabulation was done on the phytochemical content and antioxidant properties of aqueous, ethanol and methanol extracts of *C. ternatea* in Table 1 as a comparison to our study. Extraction methods are normally performed to obtain bioactive compounds from plant material. The extraction of bioactive compounds from medicinal plants is performed in many ways, including maceration, percolation, steam distillation and Soxhlet extraction, with the most common method being solvent extraction (*Zhang, Lin & Ye, 2018*). However, these conventional extraction methods need large volumes of solvents and are time-consuming and more expensive (*Wen et al., 2018*). Hence, some researchers have employed other techniques such as microwaves and ultrasounds during the solvent extraction process to reduce solvent usage, cost and extraction time while maintaining high productivity and selectivity (*Chotphruethipong, Benjakul & Kijroongrojana, 2019*). Solvent extraction involves several stages, starting with the addition of solvent into plant material. Next, the plant material will be soaked in the solvent. The extract containing the desired

components will be then filtered and collected for further usage (*Zhang, Lin & Ye, 2018*). Several factors can affect the extraction process, such as the type of solvent, the size of plant material, the extraction temperature, pH and duration time. Thus, it is crucial to select the most appropriate condition for the extraction of *C. ternatea* flowers to obtain the highest extraction yield and improve the overall extraction quality.

Most of the researchers used fresh (*Kaisoon et al., 2011*) or dried *C. ternatea* flowers (*Kamkaen & Wilkinson, 2009*; *Nithianantham et al., 2013*; *Phrueksanan, Yibchok-anun & Adisakwattana, 2014*) in their studies for extraction. To increase the extraction efficiency, the size of butterfly pea flowers was generally reduced by chopping into smaller pieces (*Havananda & Luengwilai, 2019*; *Wong & Tan, 2020*) or grinding into powdered form (*Adisakwattana et al., 2012*; *López Prado et al., 2019*; *Mehla et al., 2013*; *Verma, Itankar & Arora, 2013*). Better results were achieved when the particle size was decreased due to the increased surface area and easier penetration of solvent into the plant materials. Higher temperature was also used in several studies to enhance the solubility and diffusion rate (*Chayaratanasin et al., 2015*; *Iamsaard et al., 2014*; *Siti Azima, Noriham & Manshoor, 2017*). Although higher temperatures can enhance the efficiency of extraction, temperatures exceeding 80 °C might lead to an increased degradation rate of phytochemicals (*Salacheep et al., 2020*). In addition, the ratio between solvent and sample can also influence the extraction outcome. Normally, a higher liquid-to-solid ratio results in a higher extraction yield. However, if the ratio of liquid to solid is too high, the solvent would be too abundant, thus increasing the time needed to concentrate the extract (*Zhang, Lin & Ye, 2018*).

One of the most important steps in the extraction procedure is the type of solvent used. Generally, the selection of solvent type is determined by the compound to be extracted from the plant. Polar solvents like water and alcohol are chosen during the extraction of polar components, while non-polar solvents like hexane and chloroform are selected when extracting non-polar components (*Abubakar & Haque, 2020*). The polarity of the solvent should be similar to the polarity of the solute because this can increase the extraction efficiency. Furthermore, during solvent selection, the solubility, selectivity, safety, and cost of the solvent should also be considered as it can affect the performance of the extraction.

Solvent extraction is often utilised when isolating phenolic compounds, flavonoids, and antioxidants (*Lim, Aroua & Gew, 2021*). The isolation of these compounds is highly influenced by the type of solvent (*Barchan et al., 2014*). For instance, solvents that are widely used in the extraction of *C. ternatea* flowers are polar solvents such as water, methanol, and ethanol. This is because water, methanol and ethanol extract usually contain higher levels of phytochemicals, as well as exert stronger antioxidant abilities compared to non-polar solvent extracts. Furthermore, this flower is often used in food and beverage preparations. However, some non-polar solvents like chloroform are also used in a few studies. As discussed below, different types of solvents have been used for *C. ternatea* flower extraction.

## Water extracts of *C. ternatea* flower

The universal solvent, water is normally selected for the extraction of *C. ternatea* flowers due to its high polarity. As it is one of the most polar solvents, it can be used to extract many different polar compounds (*Abubakar & Haque, 2020*).

Based on previous studies, the total phenolic content (TPC) of the water extracts of *C. ternatea* flowers were between a wide range of 0.273 to 233.33 mg GAE/g (*Adisakwattana et al., 2012*; *Aditiyarini & Iswuryani, 2021*; *Apostolakis, Grigorakis & Makris, 2014*; *Chayaratanasin et al., 2015*; *Escher et al., 2020*; *Kamkaen & Wilkinson, 2009*; *Lakshan et al., 2019*; *López Prado et al., 2019*; *Mehla et al., 2013*; *Mehmood et al., 2019*; *Phrueksanan, Yibchok-anun & Adisakwattana, 2014*; *Salacheep et al., 2020*; *Siti Azima, Noriham & Manshoor, 2017*; *Wong & Tan, 2020*). Meanwhile, the total flavonoid content (TFC) of water extracts of *C. ternatea* flowers ranged from 6.21 to 78.28 mg QE/g (*Adisakwattana et al., 2012*; *Escher et al., 2020*; *Lakshan et al., 2019*; *Mehmood et al., 2019*; *Siti Azima, Noriham & Manshoor, 2017*). Plants normally grow according to their environmental conditions such as moisture content, light and soil aeration. Thus, different sampling locations of *C. ternatea* flowers could cause inconsistency in the TPC and TFC values. The dissimilarity in results could also be attributed to the different extraction protocols and drying procedures used in each study.

Besides flavonoids and phenolic compounds, several other studies extracted the anthocyanins of *C. ternatea* flowers using water extraction. The total anthocyanin contents (TAC) of the aqueous extractions were around the range of 1.46 to 6.93 mg CGE/g (*Chayaratanasin et al., 2015*; *Escher et al., 2020*; *Phrueksanan, Yibchok-anun & Adisakwattana, 2014*; *Salacheep et al., 2020*; *Siti Azima, Noriham & Manshoor, 2017*). The findings of TAC for these past literatures were quite consistent, which was drastically different from the huge variation seen in TPC and TFC results.

Since *C. ternatea* flowers have been proven to contain powerful antioxidant activity, many studies have performed antioxidant assays like 2,2-diphenyl-1-picrylhydrazyl (DPPH) radical scavenging, 2,2′-azino-bis (3-ethylbenzthiazoline-6-sulphonic acid) (ABTS) radical scavenging, and ferric reducing antioxidant power (FRAP) on the water extracts of these flowers to determine their antioxidant activities. Previous studies have found that the concentration needed to reduce 50% of the DPPH radical activity ($IC_{50}$) of water extracts was always higher than ascorbic acid (Vitamin C) and Trolox (*Chayaratanasin et al., 2015*; *Iamsaard et al., 2014*; *Kamkaen & Wilkinson, 2009*; *Phrueksanan, Yibchok-anun & Adisakwattana, 2014*; *Siti Azima, Noriham & Manshoor, 2017*; *Zainol et al., 2020*). The high $IC_{50}$ values indicate that the water extracts had weaker antioxidant activity when compared to the positive standards. Interestingly, *Wong & Tan (2020)* revealed that the DPPH inhibition activity of aqueous *C. ternatea* flower extract in their study was 69.22%, which was significantly higher than the 37.18% obtained from ascorbic acid (*Wong & Tan, 2020*). Besides DPPH assays, other antioxidant assays also have proven that *C. ternatea* flowers contain strong antioxidant properties (*Chayaratanasin et al., 2015*; *Escher et al., 2020*; *Iamsaard et al., 2014*; *Lakshan et al., 2019*; *Mehla et al., 2013*; *Mehmood et al., 2019*; *Siti Azima, Noriham & Manshoor, 2017*; *Wong & Tan, 2020*).

Several studies investigated the effects of different experiment parameters like the extraction temperature, period, pH, and liquid-to-solid ratio on the water extraction of dried *C. ternatea* flowers (*Aditiyarini & Iswuryani, 2021*; *Escher et al., 2020*; *Lakshan et al., 2019*). By comparing the results obtained from the various parameters, the most suitable condition for the water extraction of desired phytochemicals and antioxidants can be identified. *Lakshan et al. (2019)* reported that the optimum extraction condition was 59.6 °C at 37 min, with a flower-to-water ratio of 3 g/L (*Lakshan et al., 2019*). On the other hand, *Escher et al. (2020)* revealed that the highest amount of phenolic and anthocyanins was extracted at 40 °C and 30 min, whereas the maximum TFC and DPPH radical inhibition activity was obtained at 60 °C and 45 min. Based on statistical analysis, the lowest variability was also seen at 40 °C and 30 min, hence it was chosen as the optimum condition for the aqueous extraction of *C. ternatea* flowers (*Escher et al., 2020*). In another study, the highest TPC was achieved at 90 min, pH 7.0, and a sample-to-solvent ratio of 1 to 50 g/mL, while the maximum TAC was obtained at 90 min, pH 1.0, and a sample-to-solvent ratio of 1 to 20 g/mL (*Aditiyarini & Iswuryani, 2021*).

Ultrasound-assisted extraction (UAE) has been revealed to be an effective method for extraction due to the shorter extraction time, lower solvent usage and reduced cost (*Chotphruethipong, Benjakul & Kijroongrojana, 2019*). *Salacheep et al. (2020)* utilised the Taguchi method and grey relational analysis to determine the optimum condition for UAE of *C. ternatea* petals. High phytochemical yield, especially for TPC, was observed using a 10 mL distilled water/mg sample, at a temperature of 40 °C and extraction time of 30 min (*Salacheep et al., 2020*). A study by *Mehmood et al. (2019)* compared the difference in extraction efficiency between ultrasound-assisted extraction and non-ultrasound-assisted extraction of *C. ternatea* flowers. The water extract with ultrasound assistance showed higher amounts of phenolic and flavonoid compounds when compared to the water extract without ultrasound assistance. Similarly, stronger antioxidant activity was also seen in the DPPH, ABTS and FRAP assays of the ultrasound-assisted water extract (*Mehla et al., 2013*).

Another study by *Zainol et al. (2020)* explored the physiochemical characteristics of *C. ternatea* flowers that are microencapsulated in various amounts of gum Arabic (*Zainol et al., 2020*). Distilled water was used to extract the *C. ternatea* flowers before the microencapsulation process. Based on the results, the TPC, TFC and antioxidant activity of the samples were influenced by the amount of microencapsulation agent used. All samples were proven to exhibit antioxidative activities, with 8% gum Arabic showing the strongest antioxidant activity in both the DPPH assay and thiobarbituric acid (TBA) test. Similarly, the highest total flavonoid content was also demonstrated in the flower sample encapsulated with 8% gum Arabic. Despite that, the sample with 6% gum Arabic was seen to have the lowest TPC value among the others, whereas the sample with 2% gum Arabic was the highest. Although stronger antioxidant property was observed in samples with high flavonoid content, a good correlation was not found between the TPC and TFC results.

### Ethanol extracts of *C. ternatea* flower

Organic solvents such as alcohols are commonly used to extract bioactive components of plant materials. Ethanol is a general solvent used frequently in the extraction of *C. ternatea* flowers because of its low toxicity and high polarity. A recent study by *Tuan Putra et al. (2021)* reported that the TPC calculated at 280 nm was 102.37 mg GAE/g, which was about three times higher than 28.8 mg GAE/g calculated at 750 nm. This could be caused by the quantification of all compounds with at least one aromatic cyclic ring at the wavelength of 280 nm. In contrast, the TPC determined at 750 nm was more selective, resulting in lower amounts of phenolic compounds measured.

*Zakaria (2019)* compared the results between 70% ethanol extract and aqueous extract of *C. ternatea* flowers. The ethanol extract showed evidently better results in terms of TPC, DPPH and ABTS assays when compared to the water extract (*Zakaria, 2019*). The enhanced extraction yield observed from ethanol extract was most likely related to the increased cell permeability, leading to the release of intracellular compounds from the cells (*Tiwari et al., 2011*). In addition, *Pengkumsri, Kaewdoo & Leeprechanon (2019)* investigated the difference in extraction efficiency between water, HCl-water, 80% ethanol and HCl-Ethanol extraction (*Pengkumsri, Kaewdoo & Leeprechanon, 2019*). Similarly, the ethanol extract presented stronger antioxidant activity than the aqueous extract based on the ABTS and FRAP assays. The TPC results of ethanol extract were also higher than water extract. However, HCl-water and HCl-ethanol extract obtained a better yield of phenolic compounds as opposed to the non-acidified extracts. A study by *Kamkaen & Wilkinson (2009)* macerated *C. ternatea* petals in both 95% ethanol and distilled water respectively before the extraction procedure. Interestingly, the concentration needed to reduce DPPH radical activity to 50% ($IC_{50}$) for the ethanol extract was 4 mg/mL, whereas the $IC_{50}$ for the water extract was 1 mg/mL. The high $IC_{50}$ value of the ethanol extract represented a weaker reduction of DPPH radical activity than the water extract (*Kamkaen & Wilkinson, 2009*). This discrepancy might be related to the different procedures used during the extraction of *C. ternatea* flowers as well as the different sample species.

*Zagórska-Dziok et al. (2021)* determined the antioxidant abilities of *C. ternatea* flowers by using a water and ethanol mixture (80:20 v/v) with ultrasound assistance. The findings from the DPPH and ABTS radical scavenging assays showed that the flowers indeed contained antioxidant properties. Maximum scavenging activity was observed at the extract concentration of 1,000 µg/mL, indicating that the amount of reduction in radicals was dose-dependent *Zagórska-Dziok et al. (2021)*. Higher concentrations of extract carry more biologically active compounds, which can remove more free radicals and enhance the antioxidant properties.

In microwave-assisted extraction (MAE), the energy from microwaves aids in the penetration of solvent into the sample and enhances the movement of solutes into the solvent by interfering with the hydrogen bond. This phenomenon can occur due to the electromagnetic waves produced, which heat the sample up by promoting dipole rotation of the molecules (*Kaufmann & Christen, 2002*). For example, *Saejung, Don-In & Chimsook (2021)* extracted whole *C. ternatea* flowers in 95% ethanol with the use of microwaves. The microwave power set for this study was 400 W, with an extraction duration of 3 min,

and a liquid-to-solid ratio of 20:1 (v/w). The TAC measured from the ethanolic extract of this study was 0.2546 mg/g (*Saejung, Don-In & Chimsook, 2021*). A past study by *Izirwan et al. (2020)* also applied MAE during the extraction of *C. ternatea* using 95% ethanol (*Izirwan et al., 2020*). The maximum amount of anthocyanin extracted was 0.4571 mg/g, which was higher than those obtained by *Saejung, Don-In & Chimsook (2021)*. This might be associated with the conditions used during the experiment procedure. Response surface methodology (RSM) was applied by *Izirwan et al. (2020)* to find out the best extraction condition for anthocyanins in 95% ethanol. By amending the extraction time, temperature, and liquid-to-solid ratio, the optimum extraction condition determined was at 60 °C, 15 min, a liquid-to-solid ratio of 15:1 with microwave assistance (*Izirwan et al., 2020*).

Furthermore, RSM was also utilised by *Jaafar, Ramli & Salleh (2020)* to discover the most appropriate extraction condition for the ethanolic extract of *C. ternatea* flowers without using microwave assistance. The optimum condition for extraction was using 36.92% of ethanol, at a temperature of 44.24 °C and an extraction time of 90 min. The variation in optimum extraction conditions for these two studies might be influenced by the presence of microwave utilization during the extraction process. Thus, MAE was proven to be more time-saving as compared to the conventional extraction method, which needed 3 min and 90 min, respectively (*Jaafar, Ramli & Salleh, 2020*).

## Methanol extracts of *C. ternatea* flower

Apart from ethanol, methanol is also an organic solvent used in a lot of plant extraction procedures. Most studies perform solvent extraction with aqueous methanol mixtures instead of just using water alone to enhance the extraction efficiency. In general, methanol increases the solubility of bioactive compounds, thereby aiding in the penetration of solvent into the plant materials during the extraction process. Reports have confirmed that methanol is an effective solvent for the extraction of phytochemicals and antioxidants (*Barchan et al., 2014*; *Yen, Wu & Duh, 1996*). Previous extraction studies on *C. ternatea* flowers using methanol or aqueous methanol had successfully isolated flavonoid and phenolic compounds, as well as demonstrated strong antioxidant activity (*Kaisoon et al., 2011*; *Mehla et al., 2013*). For instance, *Nithianantham et al. (2013)* used 100% methanol for the extraction of *C. ternatea* flowers. Interestingly, the methanolic extract had similar antioxidant activity as the positive control, Butylated hydroxyl toluene (BHT) in the DPPH assay. It also exerted higher antioxidant activity than vitamin E, a commercial antioxidant (*Nithianantham et al., 2013*).

*Rabeta & An Nabil (2013)* compared the extraction efficiency of water extract and 70% methanol extract. The aqueous methanol was found to contain 3 times more phenolic content when compared with the water extract (*Rabeta & An Nabil, 2013*). However, the extraction temperature and duration were not consistent between both extractions, which could be the factors affecting the overall results of the experiment. Hence, the experiment variables should be fixed for better comparison and justification between the results obtained.

Another study by *Mehla et al. (2013)* compared the effects of extraction between the water extract and 50% methanolic extract of *C. ternatea*. The hydroalcoholic extract had

higher TPC and TFC values as opposed to the aqueous extract. Besides that, the 50% methanolic extract also showed stronger antioxidant activity in all antioxidant assays, making it a more suitable candidate for extractions (*Mehla et al., 2013*). On the other hand, *López Prado et al. (2019)* compared between water, 50% methanol and 100% methanol extract of *C. ternatea* petals with varying soaking times. They reported that the most appropriate soaking time for practical application was 6 h. According to the phytochemical yields and antioxidant activity, the water extract and 50% methanol extract achieved equivalent or better extraction outcomes than the 100% methanol extract (*López Prado et al., 2019*). As mentioned previously, the difference in results could be caused by the variation in flower origin, extraction condition, or quantification technique.

*Havananda & Luengwilai (2019)* investigated the phytochemical and antioxidant properties of 46 different *C. ternatea* accessions with different origins, flower colour and flower types using 99.95% methanol. Among all the assays performed, only TAC was reported to have a significant difference between the accessions. It was proven that the anthocyanin content was associated with the petal colour, with white petals producing lesser anthocyanins than blue/purple petals (*Havananda & Luengwilai, 2019*).

## Others solvent extracts *C. ternatea* flower

*Siti Azima, Noriham & Manshoor (2017)* prepared the *C. ternatea* flower extraction using 100 mM citrate buffer as solvent. According to the DPPH assay, the efficient concentration to scavenge 50% of the DPPH free radical ($EC_{50}$) was $0.49 \pm 0.01$ mg/mL, which was higher than the ascorbic acid ($0.12 \pm 0.00$ mg/mL) and BHA/BHT combination ($0.10 \pm 0.00$ mg/mL). The high $EC_{50}$ value indicated that the citrate buffer extract contained weaker antioxidant capacity when compared to the standards (*Siti Azima, Noriham & Manshoor, 2017*).

*Rajamanickam, Kalaivanan & Sivagnanam (2015)* studied the difference between three different types of solvent extractions, which are chloroform extract, ethyl acetate extract and 95% methanol extract of *C. ternatea* flowers. The flowers were initially extracted using 95% methanol. To obtain the crude extract of chloroform and ethyl acetate, the residue from the methanolic extract was placed in hot water before partition with chloroform and ethyl acetate respectively. The DPPH radical scavenging effect of the chloroform extract was measured to be the highest, achieving $92.46 \pm 0.05\%$ inhibition at the concentration of 1.0 mg/mL. The ethyl acetate extract obtained an inhibition of $88.79 \pm 0.20\%$ at 1.0 mg/mL, which was slightly lower than the chloroform extract. The lowest DPPH radical scavenging activity was observed in 95% methanol extract, dropping down to $85.27 \pm 0.02\%$ inhibition. Despite the low DPPH radical scavenging ability, the concentration of 95% methanolic extract required to reduce 50% of the DPPH radical activity ($IC_{50}$) was the closest with the antioxidant standard (ascorbic acid) used in the experiment, followed by ethyl acetate and chloroform extract (*Rajamanickam, Kalaivanan & Sivagnanam, 2015*).

## MATERIALS AND METHODS

### Materials

Dried butterfly pea flowers (without petals) were purchased from Jonker Street, Melaka, Malaysia. Glycerol, aluminium chloride (98%, w/w), 2,2-diphenyl-1-picrylhydrazyl (DPPH), Folin–Ciocalteu reagent, methanol, sodium carbonate, sodium acetate and the standard compounds such as ascorbic acid (99%, w/w), gallic acid and quercetin (HPLC grade) were purchased from Sigma-Aldrich. Ethanol (99.5%, w/w) was purchased from Systerm, Shah Alam, Malaysia.

### Extraction of *C. Ternatea* flower

*C. ternatea* extracts (0.04 g/ml) were prepared using various solvents including: ethanol, water, glycerol, and glycerol/water (ratio of 1:1). These extracts of *C. ternatea* were obtained through solvent extraction of the dried butterfly pea flowers (1 g), with 25 ml of the respective solvent. *C. ternatea* extracts were incubated at room temperature (27 °C) for 72 hr. Sonication (15 min, 27 °C) was conducted every 24 h during the incubation period. Following incubation, extract samples were centrifuged (2,000 rpm, 10 min) and filtered. Ethanol extract (EE) and water extract (WE) were filtered through 0.45 µm and 0.22 µm nylon syringe membrane filters, whereas 0.22 µm mixed cellulose ester syringe membrane filters were used to filter glycerol extract (GE) and glycerol/water extract (GWE). The extracts were stored in the refrigerator (4 °C) until further use. The extraction procedure was performed in triplicates.

### Determination of total phenolic content (TPC)

TPC of *C. ternatea* extracts were determined following the Folin-Ciocalteu method as described by *Mazzucotelli et al. (2018)*, with slight modification (*Mazzucotelli et al., 2018*). All extract samples were diluted with their respective extraction solvent to a concentration of 10 mg/mL. Briefly, the extract sample (5 µL), Folin-Ciocalteu reagent (25 µL), 20% sodium carbonate (75 µL), and double-distilled water (45 µL) were added and mixed by vortex in the listed order for every extract sample in a reaction tube. Samples (100 µL) were then transferred in triplicates into a 96-well plate following incubation in the dark for 60 min. The absorbance was measured at 750 nm using a microplate reader (Infinite® 200 PRO; Tecan Trading AG, Männedorf, Switzerland). TPC was calculated with a standard curve prepared using gallic acid as standard, under the same conditions as the extract samples. The results were expressed as mg of gallic acid equivalents (mg GAE g$^{-1}$).

### Determination of total flavonoid content (TFC)

The TFC of *C. ternatea* extracts was determined following the aluminium chloride colorimetric method described by *El-Guendouz et al. (2016)*, with some modifications (*El-Guendouz et al., 2016*). All extract samples were diluted with their respective extraction solvent to a concentration of 30 mg/mL. Briefly, extract sample (10 µL), 2% aluminium chloride (250 µL), 1M sodium acetate (250 µL), and double-distilled water (490 µL) were added and mixed by vortex for every extract sample in a reaction tube, followed by incubation for 15 min. Following incubation, the absorbance was measured at 425 nm using

a microplate reader (Infinite® 200 PRO; Tecan Trading AG, Männedorf, Switzerland). TFC of *C. ternatea* extracts were calculated with a standard curve prepared using quercetin as standard, under the same conditions as the extract samples. The results were expressed as mg of quercetin equivalents (mg QE g$^{-1}$).

### DPPH assay

DPPH scavenging activity (%) of *C. ternatea* extracts was determined following the method described by *Sridhar & Charles (2019)* with slight modifications (*Sridhar & Charles, 2019*). For the DPPH assay, *C. ternatea* extracts were diluted six times in a two-fold dilution method starting from 10 mg/mL, including the preparation of a blank. The same dilution procedure was repeated for standard (ascorbic acid), starting from 25 µg/mL. 50 µL of extract sample or standard (ascorbic acid) was added to the same volume of 0.1 mM DPPH methanolic solution (1,000 µL). Mixtures were vortexed and centrifuged at 6,000 rpm for 3 min, followed by incubation in the dark for 30 min. A decrease in absorbance was measured at 518 nm against a blank set of methanol without DPPH, using a microplate reader (Infinite® 200 PRO; Tecan Trading AG, Männedorf, Switzerland). The capability of *C. ternatea* extracts in reducing DPPH was determined using the following Eq. (1):

$$\text{Free radical scavenging activity}(\%) = \left( \frac{A_{\text{control}} - A_{\text{sample}}}{A_{\text{control}}} \right) \times 100\% \qquad (1)$$

where $A_{\text{control}}$ is the absorbance of the control (blank set) and $A_{\text{sample}}$ is the absorbance of the *C. ternatea* extracts. A graph of DPPH scavenging activity (%) *vs.* concentration of samples was plotted (Fig. 1) and the EC$_{50}$ values were calculated using the equation obtained from the graph. All antioxidant assays were performed in triplicates.

### Statistical analysis

The means of data of the antioxidant profile and activity of *C. ternatea* extracts were subjected to one-way analysis of variance (ANOVA) with *post-hoc* tests. The test was used to analyse the data of TPC, while Tamhane's t2 test was used to analyse the data of TFC and DPPH assay. The software used for statistical analysis was the IBM SPSS statistics software version 27 (SPSS Inc., Chicago, IL, USA).

## RESULTS AND DISCUSSION

### Antioxidant profile & activity of *C. ternatea* extracts

The phenolic compounds presence in the extracted samples were quantified through the Folin-Ciocalteu assay and the results are derived from a calibration curve that utilised gallic acid or caffeic acid as the standard, reported in gallic acid equivalents (GAE) and caffeic acid equivalents (CAE) per gram dry extract weight, respectively. For the determination of total flavonoid content (TFC) in natural extracts, several authors performed an aluminium chloride colorimetric assay by constructing a calibration curve that utilised standard solutions such as rutin and catechin with various concentrations, which the results are expressed as mg rutin (RtE) and catechin (CEQ) equivalents per gram dry weight respectively.

**Table 3** **The phytochemical content (TPC and TFC) and antioxidant activity DPPH (EC$_{50}$) of *C. ternatea* extracts superficial lowercase letter "a" represent the significant difference between BFP extracts at *p*<0.05 significant level.**

| Extracts | Phytochemical content | | DPPH |
|---|---|---|---|
| | TPC (mg GAE g$^{-1}$) | TFC (mg QE g$^{-1}$) | EC$_{50}$ (mg/mL) |
| EE | 18.6408 ± 2.5845[a] | 3.6834 ± 0.0643[a] | 14.0362 ± 1.3233[a] |
| WE | 9.8630 ± 2.3702[a] | 2.5130 ± 0.2443[a] | 6.0604 ± 0.2138[a] |
| GE | 18.8000 ± 2.3312[a] | 5.6204 ± 1.4401 | 5.4054 ± 0.0239[a] |
| GWE | 18.6259 ± 2.4509[a] | 5.8093 ± 0.5399[a] | 5.3079 ± 0.1183[a] |
| Ascorbic Acid | n/a | n/a | 0.0355 ± 0.0001[a] |

**Notes.**

EC$_{50}$ values of *C. ternatea* extracts were compared against ascorbic acid as standard.

For TPC extraction, the water extract (9.8630 ± 2.3702 mg GAE g$^{-1}$) was observed to be significantly lower than the other three extracts, with the highest being GE (18.8000 ± 2.3312 mg GAE g$^{-1}$) (Table 3). On the other hand, the TFC results for EE and WE were also lower than GE and GWE, with GWE obtaining the highest amount (5.8093 ± 0.5399 mg QE g$^{-1}$). Overall, both GE and GWE have demonstrated high phenolic and flavonoid profiles as compared to EE and WE. These findings evidently showed that glycerol poses a great ability to extract phenolics and flavonoids from *C. ternatea* as compared to ethanol and water. This could be explained by the polar structure of flavanols and polyphenols. Flavanols are relatively polar due to the presence of carbonyl and hydroxyl groups in their chemical structure (*Huamán-Castilla et al., 2020*); hence, it is expected that glycerol and glycerol/water solvent which are both polar, would perform a better recovery rate of flavonoids when compared to ethanol and water solvents. In addition, the findings from TPC and TFC results further confirmed the presence of phenolics and flavonoids in the tested *C. ternatea* extracts, and their respective contributions to the antioxidant capacity of *C. ternatea* extracts. Previous studies have mentioned that the antioxidant activities of plant extracts are highly related to their phenolic contents, such that a high phenolic profile indicates high antioxidant capacity (*Gülçin et al., 2010*). The antioxidant properties of phenolic compounds are mainly contributed by the donation of H atoms from their aromatic OH groups, hence promoting their ability to scavenge free radicals (*Yi et al., 2020*). With that being said, GWE and GE which contain the highest phenolic profiles are expected to exhibit the highest antioxidant activity as compared to EE and WE.

As seen from previous works of literature, the TPC values are higher than the amount of TFC regardless of the extraction solvents used (*Chayaratanasin et al., 2015*; *Lakshan et al., 2019*; *Mehmood et al., 2019*; *Nithianantham et al., 2013*). In this study, the amount of TPC obtained for all extracts was higher than their respective TFC values. This finding corroborated with many of the previous studies. It may be justified that the higher value of TPC obtained compared to the TFC is because flavonoid is considered one of the phenolic groups (*Ahmad et al., 2020*). Therefore, the results might be inaccurate if the TFC extracted is higher than the TPC. Interestingly, the TPC and TFC results obtained from this study were higher than in past works of literature. For example, the TPC for 70% ethanol (53 mg GAE g$^{-1}$) and water extracts of *C. ternatea* flowers (38.5 mg GAE g$^{-1}$) were noticeably

lower in the study by *Zakaria (2019)* (*Zakaria, 2019*). In another study, the TPC for 37% ethanol extract of *C. ternatea* at 45 °C and 90 min was revealed to be 41.17 ± 0.5 mg GAE g$^{-1}$, which was significantly lower than the TPC for EE in this study (*Jaafar, Ramli & Salleh, 2020*). However, the TFC for their ethanol extract (187.05 ± 3.18 mg QE/g$^{-1}$) was higher than our TFC for EE. Some discrepancy was seen between the results obtained in this study and the results obtained from previous literature. The inconsistency in TPC and TFC values might be influenced by the concentration of solvent used. Besides that, the TPC and TFC examined by various extraction methods might lead to output differences (*Katsube et al., 2004*).

### Radical scavenging activity of *C. ternatea* extracts

The antioxidant activity of *C. ternatea* extracts was determined as radical scavenging activities (%) using DPPH assay. As shown in Fig. 2, the percentage of radical scavenging activity of *C. ternatea* extracts increased with increasing concentration of the extracts. This showed that the free DPPH radicals were removed by the *C. ternatea* extracts in a concentration-dependent manner. Higher concentrations of *C. ternatea* extracts exhibited stronger antioxidant power in contrast to lower extract concentrations. Among the four *C. ternatea* extracts, GWE exhibited the highest scavenging activity (86.0713 ± 0.5743%), followed by GE (84.2893 ± 1.4676%), WE (75.3110 ± 1.1977%), and EE (36.1405 ± 4.0331%). These results confirmed that GWE and GE of *C. ternatea* demonstrated superior levels of reducing activities when compared to WE and EE, which showed a similar trend with the TPC and TFC results. As mentioned by *Gülçin et al. (2010)*, the antioxidant properties of the extracts are associated with the quantity of phenolic compounds contained in the respective extracts. Therefore, the high phenolic contents of GE and GWE might have contributed to their powerful antioxidant activities.

The antioxidant capacity of extracts was evaluated in radical scavenging power (%) through DPPH assay. DPPH are stable free radicals that can be converted to the non-radical form DPPH-H upon acceptance of an electron (hydrogen) in the presence of an antioxidant agent, this reduction is accompanied by a fade in DPPH's intense violet colour or a conversion to yellow to colourless solution. The scavenging power of *C. ternatea* extracts was hence examined by measuring the decrease in absorbance upon reduction of DPPH. In plant extracts, phenolic compounds usually donate H atoms from their aromatic OH groups to the free DPPH radicals, thereby converting them into non-radical forms. Consequently, higher amounts of phenolic content result in a higher ability to scavenge free DPPH radicals. GWE and GE with high concentrations of TPC and TFC have exhibited excellent levels of reducing activities when compared to EE and WE. Therefore, GWE and GE can potentially be used to neutralise the free radicals present in food preparation or cosmeceutical products, which decreases the chance of ageing and illnesses in humans.

The EC$_{50}$ values of *C. ternatea* extracts and the standard (ascorbic acid) were also calculated graphically and presented in Table 3. These values represent the concentration of sample needed to decrease DPPH concentration by 50%. In general, EC$_{50}$ values are inversely proportional to the free radical scavenging activity of the plant extracts. The lower the EC$_{50}$ values, the higher the antioxidant capacity of the tested extract (*Li, Wu &*

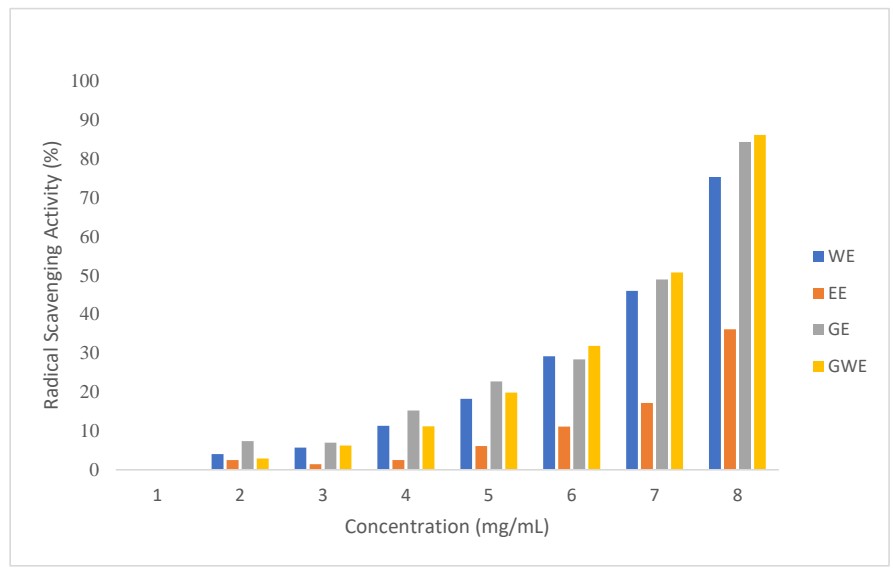

**Figure 2** **DPPH radical scavenging activity of *C. ternatea* extracts.** Blue = WE, Orange = EE, Grey = GE, Yellow = GWE.

*Huang, 2009*). Our results (in Table 3) showed that GE and GWE have exhibited the lowest $EC_{50}$ values, indicating that they possess the highest antioxidant activity as compared to WE and EE. EE, on the other hand, have shown the highest $EC_{50}$ value, indicating its weak antioxidant property. A number of studies have performed DPPH assays on *C. ternatea* flowers of various extraction solvents to assess the antioxidant activity of these flowers. For instance, the $IC_{50}$ for 70% ethanol and water extract were 0.1889 mg/mL and 0.1955 mg/mL, respectively (*Zakaria, 2019*). The findings were significantly lower than the results obtained in this study, which suggested that their ethanol and water extracts had stronger antioxidant activity than our EE and WE. On the other hand, the concentration to reduce 50% DPPH radical activity for water (1 mg/mL) was lower than 100% ethanol (4 mg/mL) in the study by *Kamkaen & Wilkinson (2009)*. The results aligned with our results, indicating that the water extract was more potent than the ethanol extract and had a higher antioxidant capacity. According to other studies that have investigated the antioxidant activity of *C. ternatea* water extracts, all their $IC_{50}$ were remarkably lower, ranging from $0.084 \pm 0.002$ mg/mL to $0.470 \pm 0.010$ mg/mL (*Chayaratanasin et al., 2015*; *Iamsaard et al., 2014*; *Phrueksanan, Yibchok-anun & Adisakwattana, 2014*). When compared to previous literature, the antioxidant activity measured by DPPH assays in this study was significantly weaker. The higher concentration required to remove 50% of DPPH radical might be affected by the temperature and time used during the extraction process.

## CONCLUSIONS

The exploration of the use of glycerol in natural product extraction will reduce the use of hazardous organic solvents and promote zero-waste biorefinery. This impact is in line with 12 Principles of Green Chemistry and Sustainable Development Goal (SDG) 12, where we perform extraction using non-hazardous chemicals and zero management of chemical waste is required. Furthermore, the reduction of utilization of volatile organic solvents may significantly reduce carbon emissions into the environment. *C. ternatea* is commonly used in food preparation, it adds a natural purplish-blue tint to the dishes without artificial food colourings. Glycerol and glycerol/water mixture were proven to possess the highest extraction efficiency when compared to other solvents including water, and the organic solvent (ethanol). This statement is supported by the high TPC and TFC profiles of GE and GWE, as well as their high radical scavenging power among the tested extracts. This study suggests glycerol as a promising extraction medium to extract higher concentrations of phytochemical contents from *C. ternatea*. Therefore, we proposed that it could be used as a natural source of antioxidant boosters, particularly, in food preparation and cosmeceutical product development.

### Funding
The authors received no funding for this work.

### Competing Interests
The authors declare there are no competing interests.

### Author Contributions
- Lai Ti Gew conceived and designed the experiments, performed the experiments, analyzed the data, performed the computation work, prepared figures and/or tables, authored or reviewed drafts of the article, and approved the final draft.
- Waye Juin Teoh performed the experiments, analyzed the data, prepared figures and/or tables, authored or reviewed drafts of the article, and approved the final draft.
- Li Lin Lein performed the experiments, analyzed the data, prepared figures and/or tables, authored or reviewed drafts of the article, and approved the final draft.
- Min Wen Lim performed the experiments, analyzed the data, performed the computation work, prepared figures and/or tables, authored or reviewed drafts of the article, and approved the final draft.
- Patrick Cognet analyzed the data, authored or reviewed drafts of the article, and approved the final draft.
- Mohamed Kheireddine Aroua conceived and designed the experiments, analyzed the data, authored or reviewed drafts of the article, and approved the final draft.

### Data Availability
The raw data is available in the Supplementary File.

## Supplemental Information

Supplemental information for this article can be found online at http://dx.doi.org/10.7717/peerj-achem.30#supplemental-information.

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
