# Peer review of "Glycerol-based extracts of *Clitoria ternatea* (Butterfly Pea Flower) with enhanced antioxidant potential"

_PeerJ Analytical Chemistry, doi:10.7717/peerj-achem.30_

## Round 0.1 · original submission · Major Revisions

One reviewer recommended reject and the other major revisions. Please verify all comments and respond accordingly.

Reviewer 1 ·

Basic reporting

The research is in the scope of the journal and it is very current as it seeks an alternative to harmful solvents for the extraction of phenolic compounds from plants.
The English language should be improved in the text. Some sentences are difficult to understand. A few suggestions are highlighted in the PDF file.
The article has a section ‘Literature Review’ that does not support this type of paper, which is a research manuscript. So I recommend shortening the subheadings about extraction methods, and using them in the ‘Introduction’ or ‘Discussion’ sections. So Table 1 should be excluded and Table 2 should be cited in the ‘Introduction’ as Table 1 because it is the first table that appear in the text. Moreover the ‘Discussion’ could be separated from the ‘Results’ section.
There are no results in the Abstract. I suggest putting some result comparing the extracts, for example the total phenolic content and the % of antioxidant activity.
The text is justified and not aligned left conform to Peer J standards.

Experimental design

The research is in the scope of the journal and it is very current as it seeks an alternative to harmful solvents for the extraction of phenolic compounds from plants.
The objectives of the research are well defined and relevant. The methods are well described.

Validity of the findings

The results are promising and show the possibility of use glycerol as an alternative to ethanol and methanol to extract polyphenolic compounds from C. ternatea.
Statistics is not discussed in the text, only cited in the methods.

Additional comments

Some suggestions and comments are highlighted in the PDF file.
Several times the two names (genus and species) of the plant are written with the first letter in capital letters. The first letter of the generic name should be always capitalized, while that of the species is not. Similarly, both parts are italicized.
Number of digits/significant figures in the results of the antioxidant assays needs to be revised.

Annotated reviews are not available for download in order to protect the identity of reviewers who chose to remain anonymous.

Reviewer 2 ·

Basic reporting

The hypothesis and background of his manuscript are not clear. And overall of the article structure, figures and tables are not good enough for a scientific presentation

Experimental design

This manuscript only presented TPC, TFC and DPPH models of different solvents, it is not a good enough scientific result to support the hypothesis of this study

Validity of the findings

The result of this manuscript is very simple result and not novelty.

Additional comments

Glycerol-based extracts of Clitoria Ternatea (Butterfly Pea Flower) with enhanced antioxidant
potential (#83750):

The authors evaluated the effectiveness of glycerol in phytochemical extraction and the antioxidant activity using the DPPH assay. Total phenolic content and total flavonoid content were used as the biomarkers in this study. A literature review of several solvent extractions was also reviewed in this study.

I have comment are;

1. All tables and figures are not good enough for a scientific presentation.
-Table 2: The author should summarize the information and show only the relevant information.

2. Abstract is not shown the result to support the summary at the end of paragraph.

3. Introduction; The first paragraph does not support the hypothesis of this study.

3. Part 2: Literature review: This section seems to bring together all the results of several studies. Authors should rewrite-summarizing important points from the previous study that relate/ or support this research.

4. Topic 2.4 and 2.5: The polarity index units of methanol (5.1) and ethanol (5.2) is quite similar, and it can be extracted from a natural product with the same phytochemical group.

5. Experimental design:
This manuscript only presented TPC, TFC and DPPH models of different solvents, it is not a good enough scientific result to support the hypothesis of this study.
C. ternarea is a well-known plant and provides both biological activity biomarker and antioxidant activity. Therefore, author should be more evaluated for bioactive compounds

Please see this study which reported the correlation between the phytochemical content and biological activity “Escher GB, Marques MB, do Carmo MA, Azevedo L, Furtado MM, Sant'Ana AS, da Silva MC, Genovese MI, Wen M, Zhang L, Oh WY. Clitoria ternatea L. petal bioactive compounds display antioxidant, antihemolytic and antihypertensive effects, inhibit α-amylase and α-glucosidase activities and reduce human LDL cholesterol and DNA induced oxidation. Food research international. 2020 Feb 1;128:108763.”

12-6-2023

---

## Round 0.2 · Minor Revisions

The English language was improved in the text by following the suggestions highlighted in the previous PDF file but I yet suggest a revision by a fluent English speaker.

**Language Note:** The Academic Editor has identified that the English language must be improved. PeerJ can provide language editing services - please contact us at [email protected] for pricing (be sure to provide your manuscript number and title). Alternatively, you should make your own arrangements to improve the language quality and provide details in your response letter. – PeerJ Staff

Reviewer 1 ·

Basic reporting

The English language was improved in the text by following the suggestions highlighted in the previous PDF file but I yet suggest a revision by a fluent English speaker.
The main observation of the study was included as a result in the abstract, concluding that glycerol extract and glycerol/water extract demonstrated higher phenolic and flavonoid contents than ethanol and water extract.
Based on the journal standards the Research Manuscript does not support a section ‘Literature Review’ but the authors decided to keep it.

Experimental design

No comment

Validity of the findings

No comment

Additional comments

The authors accepted other suggestions that were marked in the previous PDF file and the article was improved.

Reviewer 2 ·

Basic reporting

No comment

Experimental design

No comment

Validity of the findings

No comment

Additional comments

This manuscript is now able for publication in this journal

---

## Round 0.3 · accepted · Accept

The revised version is well written and the result is good enough to meet the requirements of this journal. However, please correct the small typos reported by the reviewer for this final round in the proof stage.

Reviewer 2 ·

Basic reporting

The revised version is well written and the result is good enough to meet the requirements of this journal

Experimental design

The experimental design is well designed and meets the requirements of this journal

Validity of the findings

Conclusions are well stated

Additional comments

The author should be carefully check the correction.

Some typos were still found such as;
Table 1: IC50/ED50….50 should be subscript
Table 3: Does the author test the TFC value statistic for the GE sample? please check and confirm